# Predicting Lameness in Sheep Activity Using Tri-Axial Acceleration Signals

**DOI:** 10.3390/ani8010012

**Published:** 2018-01-11

**Authors:** Jamie Barwick, David Lamb, Robin Dobos, Derek Schneider, Mitchell Welch, Mark Trotter

**Affiliations:** 1Precision Agriculture Research Group, University of New England, Armidale, NSW 2351, Australia; dlamb@une.edu.au (D.L.); robin.dobos@dpi.nsw.gov.au (R.D.); dschnei5@une.edu.au (D.S.); mwelch8@une.edu.au (M.W.); 2Sheep Cooperative Research Centre, University of New England, Armidale, NSW 2351, Australia; 3New South Wales Department of Primary Industries, Livestock Industries Centre, University of New England, Armidale, NSW 2351, Australia; 4Formerly Precision Agriculture Research Group, University of New England, Armidale, NSW 2351, Australia; m.trotter@cqu.edu.au; 5Institute for Future Farming Systems, School of Medical and Applied Sciences, Central Queensland University, Central Queensland Innovation and Research Precinct, Rockhampton, QLD 4702, Australia

**Keywords:** sheep, behavior, lameness, activity, acceleromter, on-animal sensor

## Abstract

**Simple Summary:**

Monitoring livestock farmed under extensive conditions is challenging and this is particularly difficult when observing animal behaviour at an individual level. Lameness is a disease symptom that has traditionally relied on visual inspection to detect those animals with an abnormal walking pattern. More recently, accelerometer sensors have been used in other livestock industries to detect lame animals. These devices are able to record changes in activity intensity, allowing us to differentiate between a grazing, walking, and resting animal. Using these on-animal sensors, grazing, standing, walking, and lame walking were accurately detected from an ear attached sensor. With further development, this classification algorithm could be linked with an automatic livestock monitoring system to provide real time information on individual health status, something that is practically not possible under current extensive livestock production systems.

**Abstract:**

Lameness is a clinical symptom associated with a number of sheep diseases around the world, having adverse effects on weight gain, fertility, and lamb birth weight, and increasing the risk of secondary diseases. Current methods to identify lame animals rely on labour intensive visual inspection. The aim of this current study was to determine the ability of a collar, leg, and ear attached tri-axial accelerometer to discriminate between sound and lame gait movement in sheep. Data were separated into 10 s mutually exclusive behaviour epochs and subjected to Quadratic Discriminant Analysis (QDA). Initial analysis showed the high misclassification of lame grazing events with sound grazing and standing from all deployment modes. The final classification model, which included lame walking and all sound activity classes, yielded a prediction accuracy for lame locomotion of 82%, 35%, and 87% for the ear, collar, and leg deployments, respectively. Misclassification of sound walking with lame walking within the leg accelerometer dataset highlights the superiority of an ear mode of attachment for the classification of lame gait characteristics based on time series accelerometer data.

## 1. Introduction 

Lameness is one of the most common and persistent health problems in sheep flocks around the world [1], which has resulted in it becoming a common cause of economic and welfare concern in many sheep producing countries [2]. Concerns arise due to adverse effects on both the economic and physical performance of the flock [3]. Lameness is known to be a painful condition and animals experiencing pain often deviate from their normal behaviour by altering activity (either through an increase or decrease in particular behaviours), posture, gait, appetite, and appearance [4]. Given the welfare issues and productivity losses resulting from lameness, there is a need to identify lame animals as early as possible. This is especially important in the case of contagious infections such as footrot and contagious ovine digital dermatitis (CODD), where early identification is necessary to reduce transmission and also assist in decreasing susceptibility to secondary diseases such as flystrike due to an increased lying time [3]. Arthritis is another candidate disease where early detection is advocated to improve treatment efficacy [6], currently costing the Australian sheep industry $39 million annually [5]. 

One approach for assessing pain in animals is to examine their behaviour [7]. Until recently, behavioural assessments of illness have relied on subjective clinical evaluation based on the accumulated experience of livestock handlers, with lameness conventionally being identified through the visual inspection of individual animals within a flock. Animal behaviour is frequently monitored to determine the health and wellness state of the animal and identifying lame behaviour requires analysis of the animal’s gait. There are several methods available to analyse gait characteristics including visual observation, pressure measuring devices [8,9,10,11,12,13], video signals [14,15], and accelerometer systems [16]. Whilst visual locomotion scoring has the advantage of being implemented on any farm at any given time, practical considerations such as labour resources, should be taken into account. In addition to the inconsistency and subjectivity of visual locomotion scoring systems, assessing the gait of individual sheep in a flock can be practically challenging. Also, any scoring of the entire flock only provides prevalence information for that specific moment and daily monitoring of locomotion and behaviour on farms by a trained observer is too time-consuming and costly [17]. 

The approaches used to define locomotion include the observation of stride length, duration of weight bearing on both affected and unaffected limbs, body posture, and joint movement [18]. The main disadvantage of these methods is the requirement for a visual appraisal of each animal. Utilization of technology to automatically record behaviour allows for the collection of objective values without the need for human observation and also allows a greater temporal resolution of data collection. However, systems need to be developed that allow reliable and repeatable measurements of behaviours capable of indicating animal wellness state [7,19]. Martiskainen, Järvinen [20] stated that “an automated system monitoring several detailed behaviour patterns at once would be welcome in animal production as an aid to assess health and welfare”, as measures of activity may be useful to assess overall health and welfare status [21]. Furthermore, monitoring systems must be able to recognise behavioural signatures associated with normal behaviour and with non-normal behaviour indicative of compromised health status or the onset of specific diseases [22]. 

Several authors have proposed the use of accelerometers deployed on animals for lameness prediction, with commercial systems already available in the dairy industry. Accelerometers have previously been used to detect lameness in horses [23,24,25], beef cattle [23], and dairy cows [16,20,26,27,28]. In cattle, lameness has commonly been detected via a change in daily activity or a change in recumbent posture [26,27,29,30]. Few studies have attempted to identify lameness through a difference in the acceleration signal itself. Lame and sound cows were distinguished by Pastell, Tiusanen [16] using accelerometers deployed on each of the four legs and subsequent data analysis using variance and wavelet variance for each axis along with total leg acceleration. Similarly, Blomberg [31] identified differences in the acceleration across all gait phases between normal and lame cows. Using Support Vector Machines (SVM) classification models, Martiskainen, Järvinen [20] achieved a reasonable recognition of standing (80% sensitivity, 65% precision), lying (80%, 83%), ruminating (75%, 86%), feeding (75%, 81%), walking normally (79%, 79%), and lame walking (65%, 66%) in dairy cows using collar deployed inertial sensors. 

The automatic measurement of activity related to normal and abnormal locomotion characteristics would allow for daily activity measurements, and compared to the traditional approaches, could be a better option [17]. Previous studies have validated the ability of accelerometers to consistently and reliably describe sheep behaviour patterns [32,33,34,35]. The ability of such devices to detect changes in movement symptoms related to lameness is yet to be evaluated. 

The aim of this current study was to determine the ability of a tri-axial accelerometer to discriminate between sound and lame gait movement in sheep. Noting the desirability of an ear-tag form factor and the previously used modes of deployment in other species, tri-axial accelerometers were mounted on collars, the front leg, and on an ear-tag. 

## 2. Materials & Methods

### 2.1. Study Site and Animals

This study was conducted at the University of New England’s SMART Farm, Armidale, NSW, Australia (Longitude 151°35′40″ E, Latitude 30°26′09″ S). All animal experimental procedures were approved under the University of New England Animal Ethics Committee, AEC14-066.

A group of ten Merino cross Poll Dorset ewes, approximately 11 months of age with an average weight of 62 kg, were used in the present study. A subset of five animals were selected at random for instrumentation, with the remainder retained as companion animals. 

### 2.2. Instrumentation

Accelerometers (GCDC X16-mini, Gulf Coast Data Concepts, MS, USA) configured to collect signals at 12 Hz (12 samples/second) were attached simultaneously to three locations on each candidate sheep: a neck collar, the anterior side of the nearside front shin, and the ventral side of the offside ear. Each sensor was 50 × 25 × 12 mm in size and weighed 17.7 g.

The location and method of fixation for the sensors is shown in Figure 1. Collar deployed accelerometers were attached to the polycarbonate case of a UNE Tracker II GPS collar [36] and placed around the sheep’s neck. Front leg mounted sensors were attached to the foreleg shin using Vetflex self-adhesive bandage, similar to a previously used method of attachment for sensors in dairy cows [37]. Ear-tag deployed sensors were attached to a trimmed management tag (Allflex) using electrical tape. Previously deployed tags were placed in the central lower half of the animals’ offside ear; consistent with normal ear tag deployment. 

The GCDC X16-mini tri-axial accelerometer measures static and dynamic acceleration along three orthogonal axes (*X*, *Y*, and *Z*). Orientations of the *X*, *Y*, and *Z* axis in this current study were dorso-ventral, lateral, and anterior-posterior, respectively (Figure 1). 

### 2.3. Observations

The general workflow of procedures is summarised in Figure 2. Data processing and analysis were conducted in Matlab (MathWorks, R2014a) and R (v3.1.2; R Core Team, 2014). 

This study was comprised of two distinct observation periods for each animal: sound and lame, referred to as Phase I and Phase II, respectively. For Phase I, a single animal was randomly selected and restrained in a small catching pen. Accelerometers were deployed, after which three animals (one instrumented and two non-instrumented) were released into a small adjacent paddock (80 m × 6 m). 

Following this period of “sound” behaviour observations, accelerometers were removed and animals were allowed to rest for approximately 20 min. The same animal was then re-instrumented with accelerometers and released with two companion animals, and the “lame” behaviour observations commenced (Phase II). A small habituation period of 10 min after instrumentation was allocated prior to the commencement of observation recordings.

Each observation session of continuous sampling lasted approximately 2 h and the process was repeated for five animals. The movement of each instrumented animal was monitored and video recorded (Panasonic HDC-SD9). Observations were classified as listed in Table 1. 

### 2.4. Lameness Simulation

To simulate lameness, the sheep’s front right leg was restrained using VETFLEX adhesive bandage, with the hoof bent backwards and tied back near the fetlock joint (Figure 3). This method allowed the leg to remain straight at the knee; however, the animal was unable to bear any weight on the restrained limb. 

### 2.5. Developing the Behaviour Classification Model

Accelerometers and video recordings were time-synchronised to obtain a behaviour annotated dataset. Accelerometer data were downloaded using proprietary software (XLR8, Gulf Coast Data Concepts, MS, USA) and exported for further annotation with corresponding behaviours. 

For both Phase I and Phase II, the annotated files were divided into 10 s epochs, with unknown and transitional behaviour epochs (based on a visual assessment of the video recorded data) being excluded from the analysis. For each 10 s epoch, fourteen movement metrics were extracted including average, maximum, and minimum acceleration (*X*-, *Y*-, and *Z*-axes); movement variation; signal magnitude area; average intensity; entropy; and energy. These metrics are listed in detail in Table 2. Data within each deployment were combined following feature extraction, creating a single file for each deployment.

Lame recordings were analysed as a mirror image of the sound behaviours, creating four behaviour classes for classification: lame walking, lame standing, lame grazing, and lame lying. Lame lying and lame standing were excluded from the analysis as these behaviours confused the classifier and were misclassified with their corresponding sound behaviours, due to similarities between the two signals (data not shown). The following six behaviour classes were available for discrimination using the QDA: lame walking, lame grazing, sound walking, sound grazing, sound standing, and sound lying (where available). 

Given the similarities between “sound” (i.e., unaffected) and “lame” grazing behaviours, the data were separated into two analyses. Analysis I included sound walking, sound standing, sound grazing, sound lying (where data were obtained), lame walking, and lame grazing. Analysis II excluded lame grazing in an attempt to reduce the level of misclassification between lame and sound grazing events. 

As this current study classified lame behaviour based on a change in the acceleration signal (creating new behaviour categories), feature selection was required to select the subset of metrics of the most importance for both Analysis I and II. Feature selection using Random Forest (RF) [39] was performed on Analysis I and Analysis II separately and the three highest ranked features were used as the discriminating metrics for the QDA classifier.

### 2.6. Feature Relative Importance and Behaviour Classification

To systematically assess the usefulness and identify the most important features for discriminating different activities, RF ranking of importance was performed. The R libraries ‘randomForest’ [42] and ‘varSelRF’ [43] were used to identify the relative importance of the fourteen features based on their Gini index, which is used to measure the error across the RF ensemble of trees. Within ‘randomForest’, *mtry* (the number of variables tried at each split, which is approximately equal to the square root of the number of variables for classification) was set at 4 and *ntree* (the number of trees in the random forest) was set at 500. 

Quadratic discriminant analysis (QDA) was used to predict the mutually exclusive behaviour categories based on the calculated features obtained from the acceleration data sets. The top three ranked metrics of most importance identified from the RF analysis were selected as the discriminating metrics in their corresponding annotated acceleration signal dataset. 

### 2.7. Model Validation

Following the development of the classification model which provided a general prediction accuracy for each behaviour, data were validated using a leave-one-out cross validation to maintain the stability of the model. This commonly used procedure consists of consecutively training the model on all but one of the records, testing it on the one dropped from the training set, and averaging the resulting scores [44]. As the predictive ability of the classification method is tested on data that were not used to train or create the model, the resulting estimates of classification accuracy are valid estimates of those that would be obtained from a true external validation [45].

From the leave-one-out cross validation, a confusion matrix was calculated, from which the sensitivity, specificity, accuracy, and precision were calculated using the following Equations (1)–(4): (1)$$\mathrm{sensitivity}=\frac{\mathrm{true}\;\mathrm{positives}}{\left(\mathrm{true}\;\mathrm{positives}+\mathrm{false}\;\mathrm{negatives}\right)}$$
(2)$$\mathrm{specificity}=\frac{\mathrm{true}\;\mathrm{negatives}}{\left(\mathrm{true}\;\mathrm{negatives}+\mathrm{false}\;\mathrm{negatives}\right)}$$
(3)$$\mathrm{accuracy}=\frac{\left(\mathrm{true}\;\mathrm{positives}+\mathrm{true}\;\mathrm{negatives}\right)}{\left(\mathrm{true}\;\mathrm{positives}+\mathrm{true}\;\mathrm{negatives}+\mathrm{false}\;\mathrm{positives}+\mathrm{false}\;\mathrm{negatives}\right)}$$
(4)$$\mathrm{precision}=\frac{\mathrm{true}\;\mathrm{positives}}{\left(\mathrm{true}\;\mathrm{positives}+\mathrm{false}\;\mathrm{positives}\right)}$$

Here, true positive (TP) is the number of instances where the behavioural state of interest was correctly classified. False negative (FN) is the number of instances where the behavioural state of interest was visually observed but was incorrectly classified as some other behaviour. False positive (FP) is the number of instances where the behavioural state of interest was incorrectly classified in the behaviour of interest and true negative (TN) is the number of instances where the behavioural state of interest was correctly classified as not being observed. 

## 3. Results & Discussion

### 3.1. Observations

No apparent adverse effects of sensor attachment on animal behaviour were observed in the current study. A summary of the quantities of data collected for each state (both “sound” and “lame” behaviours) is given in Table 3. 

### 3.2. Visual Description of Behaviour—Ear as an Example

The head position provides valuable information for the detection of different behaviour patterns [23] and preliminary analysis identified variation between sound and lame walking in the raw ear acceleration signals. Discriminating lame walking from other behaviours, predominantly sound walking, relies on the increased range of motion experienced by the sensor created from the ‘head bobbing’ action characteristic of abnormal quadruped gait movement [46,47,48]. The increased acceleration experienced by the sensor during lame walking is shown in Figure 4. In comparison with normal walking, during lame walking, the *X-*, *Y-*, and *Z-*axes record a greater acceleration amplitude, indicative of the increased swinging motion that the sensor experiences as a result of the uneven weight distribution during lame locomotion. 

### 3.3. Feature Selection

For each accelerometer position, the order of importance of metrics as determined by the mean decrease in Gini values is summarised in Table 4. Analysis I includes sound walking, sound standing, sound grazing, sound lying (where data were obtained), lame walking, and lame grazing. Analysis II excluded lame grazing, and therefore included all sound behaviours (walking, standing, grazing, and lying) and lame walking only. There is slight variation in feature importance between Analysis I and II within each deployment, particularly evident for the collar attachment. 

### 3.4. Behaviour Prediction Algorithm

#### 3.4.1. Analysis I

The results of the QDA leave-one-out cross validation prediction for each deployment method in Analysis I are shown in Table 5.

Sound grazing and walking were well predicted from all sensor deployment locations. The collar deployed sensor yielded a poor standing and lying prediction due to misclassification with ‘grazing and walking’ and with ‘walking’ behavior, respectively. A commonality across all deployments was the misclassification of lame grazing events with sound grazing. Therefore, lame grazing behavior was removed from the model.

#### 3.4.2. Analysis II

The results of the QDA leave-one-out cross validation prediction for each deployment method in Analysis II are shown in Table 6.

##### Ear

Removing lame grazing from the model improved the prediction accuracy of all remaining behaviours. Lame walking classification increased to 82%; however, some events were still misclassified as sound walking and grazing (Table 6).

The greater acceleration in all three planes exhibited during lame walking events is a reflection of the head bobbing motion observed with abnormal locomotion, thus creating a higher range in sensor motion. The uneven weight distribution in lame animals [46] further distorts the sinusoidal acceleration signal associated with sound walking activity, creating higher values for metrics measuring the intensity of movement (namely, AI and MV). This hobbling action resulting from an uneven weight distribution on the fore and hind limbs is one of the first signs of lameness identified by sheep farmers [49], highlighting the potential advantages of using accelerometer technology to aid in lameness detection. 

The agitation and discomfort associated with flicking of the head, which has previously been linked with lameness [50], was not observed in the present study. The simulated lameness system employed here did not result from the animal experiencing pain. Rather, the method of restraint simply prevented the animal from bearing any weight on the restrained limb. Future work may investigate alternate methods of lameness simulation, i.e., turpentine injection [51], as this excessive movement of the head potentially holds classification value for the detection of individuals with early signs of lameness.

##### Collar

Removing lame grazing from the model increased the prediction accuracy of lame walking but reduced the sound walking prediction accuracy (Table 6). This resulted from an increase in sound walking events being misclassified as lying. Sound grazing and standing prediction accuracies were similar across Analysis I and II; however, more lying events were misclassified as standing in Analysis II (Table 5 and Table 6). Similar to the description for cattle [52], the main lying posture signal for sheep does not differ much from their standing posture, which explains the mutual misclassifications of lying as standing in the current study. In cattle, Martiskainen, Järvinen [20] reported that the misclassification of lame walking often occurred with either standing, feeding, or sound walking (in 32% of cases) based on an SVM classification model. This misclassification of behaviours is comparable to the current study, where lame walking was misclassified either as sound walking or standing. 

There was little difference in the acceleration signatures between lame and sound walking. The lack of deviation from a normal signal suggests that the collar attachment location dampens any change in signal, which may be associated with the uneven weight distribution and head bobbing action exhibited by lame animals. Therefore, from the findings here, a collar attachment is not recommended to identify lameness behaviour in sheep.

##### Leg

Sound grazing, lying, and lame walking were well predicted; however, the sound walking prediction accuracy was substantially reduced in Analysis II, with 34 events being misclassified as lame walking. Also, sound standing was slightly reduced, with more events being misclassified as grazing (Table 6). Due to the reduced stepping action observed during grazing behaviour in lame animals, the recorded signals between these two behaviours are similar, resulting in a high misclassification rate. As the instrumented limb was bearing all the forequarter weight of the animal, a reduced amount of movement when grazing was evident.

### 3.5. Classification Algorithm Performance

The performance of the QDA classification algorithm to discriminate sound behaviours (walking, standing, lying, and grazing) and lame walking within the three different deployment locations is shown in Table 7.

An example calculation of performance metrics for the collar deployment of sound standing behaviour derived from the confusion matrix in Table 6 is:Sensitivity = 283/(283 + 13 + 2) = 95%Specificity = 241/(241 + 28) = 90%Accuracy = (283 + 241)/(298 + 269) = 92%Precision = 283/(283 + 26 + 2) = 91%

#### Ear

The overall performance of the QDA model for the ear deployed accelerometer was high. Accuracy values for all behaviours were greater than 95%. However, the unbalanced structure of the data makes this measure difficult to interpret [53] and therefore other measures of model performance are more informative. Specificity for all behaviour categories was high (>96%). Sensitivity values were good for all sound behaviours but slightly lower for lame walking, resulting from 18 lame walking events being misclassified. Similarly, the precision for lame walking was moderate (82%), as 13 behaviour events were incorrectly predicted as lame walking (Table 7). Similar to the model used by Martiskainen, Järvinen [20], this indicates that the classifier had difficulty in predicting positive cases for the lame walking category, suggesting that this behaviour pattern is easily confused with other behaviours (such as sound walking and grazing). 

#### Collar

The overall performance of the QDA classification model for the collar deployed accelerometer was poor. The performance statistics for sound grazing behaviour was high; however, the sensitivity and precision rates for all other behaviours was low. With the exception of sound grazing, the results suggest that the classifier had difficulty in predicting positive events for the majority of behaviour classes. The accuracy, sensitivity, and precision values (83%, 35%, and 35%, respectively) for lame walking differ to those reported in the earlier work of Martiskainen, Järvinen [20] in cattle who reported corresponding values of 98%, 65%, and 66%. The low performance statistics obtained in this current study indicate that a collar deployed accelerometer holds little value for classifying lame walking behaviour in sheep using the classification model employed here. Since reducing the number of lame behaviour categories to only include lame walking failed to substantially improve the classification success, other techniques need to be evaluated in an attempt to improve the prediction results.

#### Leg

The overall performance of the QDA model for the leg deployed accelerometer was moderate. The accuracy was above 85% for all behaviours. Sensitivity for lame walking was high (87%); however, the misclassification rates for sound standing and walking behaviour categories resulted in these behaviour classes having a low sensitivity value (58% and 64%, respectively). The precision for all behaviours except sound standing was high. This suggests the standing behaviour was easily confused with other behaviours, namely grazing. 

### 3.6. General Discussion

The approach used to categorise lame walking from normal behaviours in the current study is similar to that used by Martiskainen, Järvinen [20], whereby lame behaviours are added to the classifier as an additional behaviour category. Other studies conducted on dairy cattle have used a change in the proportion of behaviours (predominantly lying) to infer a lameness state [16,26,28]. This approach has shown that around 92% of cows which developed clinical lameness also had a decrease in pedometric activity of at least 15% [54]. Additionally, extreme lying times, observed through increases or decreases in the amount of time spent lying, have been shown to be predictive of lameness events in cattle [55]. Similar detail on the association between lameness and lying behaviour is lacking for sheep; however, a decrease in the activity of lame animals has been previously indicated [3]. Further validation to quantify the total lying time to detect lameness in sheep should be investigated. Using a behavioural state proportion approach, lameness could be categorised based on a change in total activity distribution.

The novel method used to simulate lameness in this study was extreme and prevented any weight bearing on the restrained limb, creating a more distinct lameness gait than what we would expect naturally. Due to lameness activities being grouped as a different class for classification, this method was preferred to create a distinct difference in the acceleration signals compared to normal behaviour. If a difference in the acceleration pattern could not be detected with an extreme gait change as investigated here, there would be little hope of detecting the progression of a mildly lame gait. Additional approaches could be investigated that would also allow for alternate methods of detection such as walking speed, distance travelled, stride length, and stride duration, which have been suggested to be valid indicators of lameness in other species [27,46]. However, a question which was not explored here, and one that may hold diagnostic value, is “when lame sheep are grazing, do they alter the vertical angle of the ‘good’ foreleg while grazing to avoid over-balancing?” For example, when the head is lowered so they can eat grass, compared to a normal (non-lame) sheep. Perhaps less tilting of the remaining leg occurs away from the vertical position, although it is unlikely that this adjustment would be detected by an ear mounted accelerometer sensor. Additionally, lame sheep may tend to kneel while grazing, therefore altering the leg accelerometer orientation, which may hold diagnostic value. Lameness simulations should also investigate the difference between having lame fore and hind limbs as this will affect the acceleration signal obtained, ultimately influencing the classifier’s ability to discriminate between sound and lame gaits.

There is little information on the ability of sheep farmers to identify lame animals or on the decision for when a sheep farmer decides to investigate and treat a lame animal [46]. A study on sheep farmers in England concluded that their estimates on lameness prevalence within flocks was sufficiently accurate [56]. Similar information is lacking for the Australian sheep industry. A valid question is therefore, “what is the economic and welfare value of detecting lameness earlier and initiating treatment sooner?” Further research is required in this area to quantitatively describe the advantages of an electronic detection system in terms of overall productivity and animal welfare benefits. 

## 4. Conclusions

The identification of animals with abnormal gait patterns could aid in the detection of many diseases which have lameness symptoms. The visual daily monitoring of locomotion or behaviour on farms is time-consuming and hence cost inefficient. The automatic measurement of lameness-related, animal-based characteristics would allow for daily measurements and could, therefore, be a better option. This current study has shown that a tri-axial accelerometer deployed in the ear-tag can successfully discriminate lame walking activity from normal grazing, standing, and walking behaviours. The collar and leg deployed accelerometers failed to successfully classify both sound and lame walking activity. This could be a function of sensor placement and the simulation not being a true representation of lameness. Further research investigating a commercially suitable accelerometer-based, automatic identification system for the onset of lameness is warranted. Additionally, future work should investigate a detection system based on changes in indicators over time, rather than on the deviation from the group mean. Gait characteristics should also account for variations in normal and abnormal patterns in terms of animal age, sex, and breed. 

## Figures and Tables

**Figure 1 animals-08-00012-f001:**
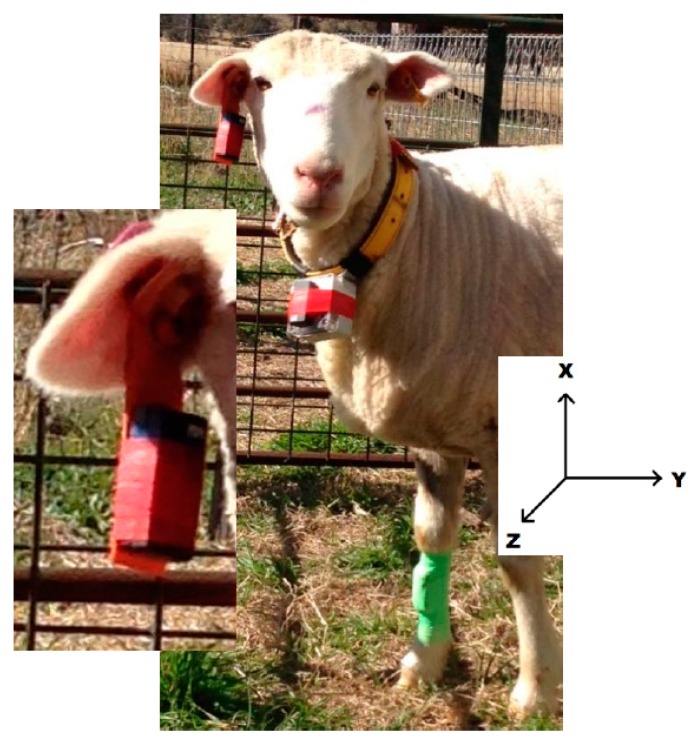
Experimental animal displaying the ear and neck (collar) locations evaluated. Insert shows the axis orientation of the sensor. Refer to Figure 3 for leg attachment display.

**Figure 2 animals-08-00012-f002:**
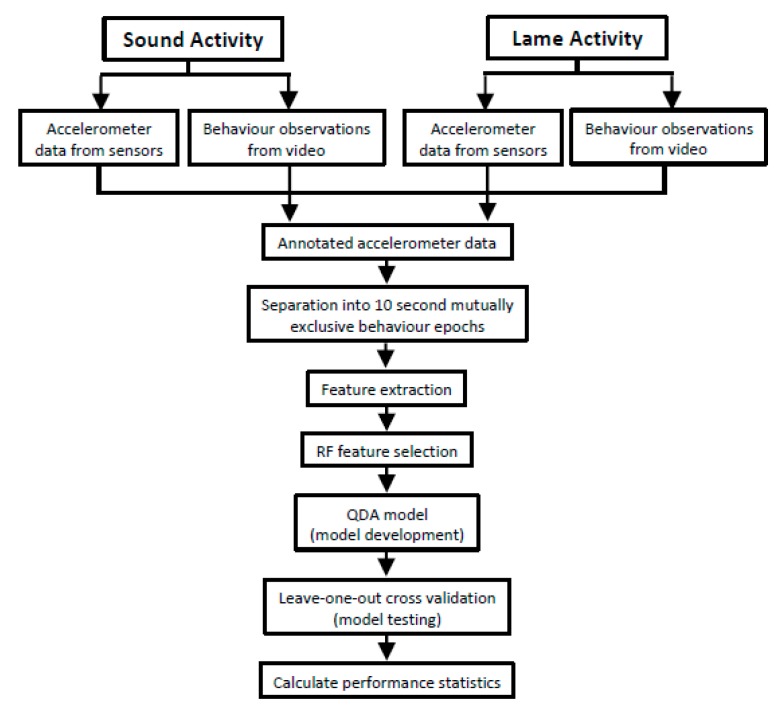
Workflow of the steps employed to classify sheep behaviour from a tri-axial accelerometer.

**Figure 3 animals-08-00012-f003:**
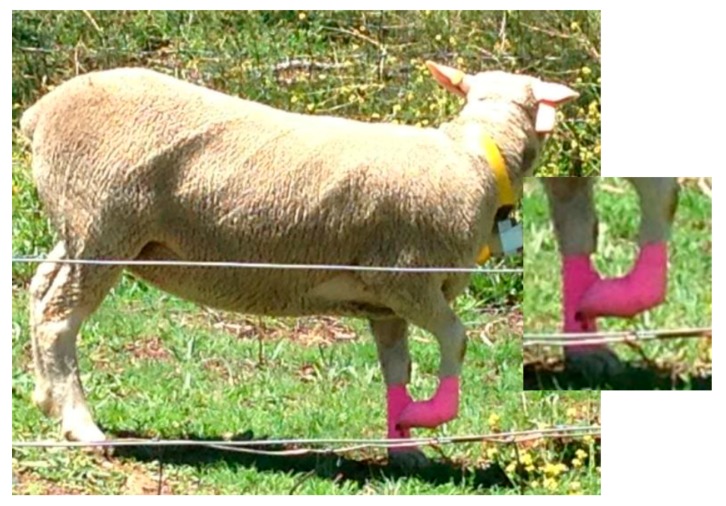
Experimental animal showing the method used to simulate lameness behaviour (main photo and inset). The method of restraint prevented any weight bearing on the restrained limb.

**Figure 4 animals-08-00012-f004:**
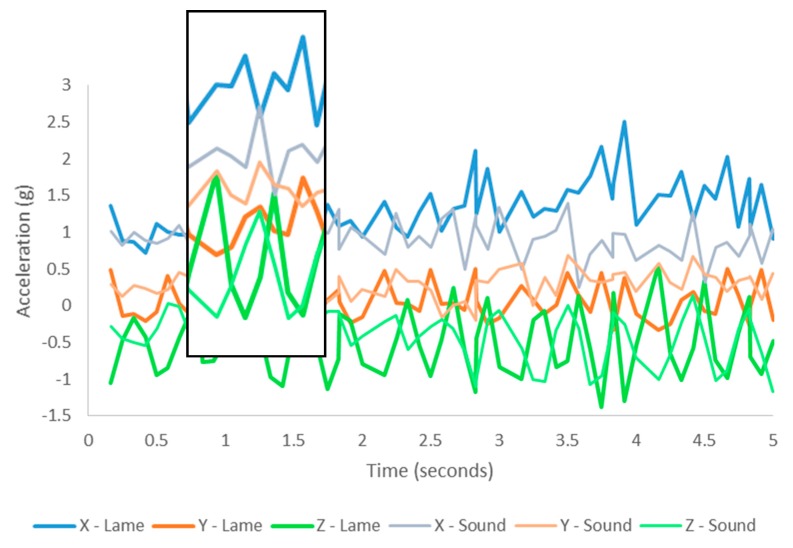
Raw ear acceleration signals for sound and lame walking. Note the increased amplitude of lame walking signals. Insert highlights the difference in amplitude between lame and sound walking signals.

**Table 1 animals-08-00012-t001:** Descriptions of the four behaviour states monitored.

Behaviour	Classification Description
Grazing	Grazing with head down or chewing with head up either standing still or moving. Rumination was classed as standing or lying.
Walking	Minimum of two progressive steps either forward/back or sideways.
Standing	Static standing with minor limb and head movements. Animal is in a standing posture whilst idle or inactive. Head may be up or down.
Lying	Animal is in a lying posture whilst idle or inactive assuming a recumbent position with minor head movements.

**Table 2 animals-08-00012-t002:** Calculated features from raw *X*, *Y*, and *Z* acceleration values.

Feature	Equation	Feature Discussed In
Average *X*-axis (Ax)	$A_{x}=\frac{1}{T}\sum_{t=1}^{T}x\left(t\right)\;$	[32]
Average *Y*-axis (Ay)	$A_{y}=\frac{1}{T}\sum_{t=1}^{T}y\left(t\right)$	[32]
Average *Z*-axis (Az)	$A_{z}=\frac{1}{T}\sum_{t=1}^{T}z\left(t\right)$	[32]
Movement Variation (MV)	$\mathrm{MV}=\frac{1}{N}\;\left(\sum_{i=1}^{N-1}\left|x_{i+1}-x_{i}\right|+\sum_{i=1}^{N-1}\left|y_{i+1}-y_{i}\right|+\sum_{i=1}^{N-1}\left|z_{i+1}-z_{i}\right|\right)$	[38]
Signal Magnitude Area (SMA)	$\mathrm{SMA}=\frac{1}{T\;}\left(\sum_{t=1}^{T}\left|a_{x}\left(t\right)\right|+\sum_{t=1}^{T}\left|a_{y}\left(t\right)\right|+\sum_{t=1}^{T}\left|a_{z}\left(t\right)\right|\right)$	[38]
Average Intensity (AI)	$\mathrm{AI}=\;\frac{1}{T}\left(\sum_{t=1}^{T}\mathrm{MI}\left(t\right)\right)$ where $\mathrm{MI}\left(t\right)=\;\sqrt{a_{x}{\left(t\right)}^{2}+{\;a}_{y}{\left(t\right)}^{2}+{\;a}_{z}{\left(t\right)}^{2}}$	[39]
Entropy	$S=\;\frac{1}{n}\sum^{\text{​}}\left(1+{\;\mathrm{Ts}}_{i}\right)\ln\left(1+{\;\mathrm{Ts}}_{i}\right)$ where n is the number of records in the burst and Ts = A_z_ + A_y_ + A_z_	[40]
Energy	$$E=\frac{1}{n}\sum \left({\mathrm{TSS}}_{i}^{2}\right)$$ where n is the number of records in the burst and TSS = A_z_^2^ + A_y_^2^ + A_z_^2^	[40]
Maximum *X* (Max*X*)	The maximum *X*-axis acceleration value within the epoch	[20,41]
Maximum *Y* (Max*Y*)	The maximum *Y*-axis acceleration value within the epoch	[20,41]
Maximum *Z* (Max*Z*)	The maximum *Z*-axis acceleration value within the epoch	[20,41]
Minimum *X* (Min*X*)	The minimum X-axis acceleration value within the epoch	[20,41]
Minimum *Y* (Min*Y*)	The minimum *Y*-axis acceleration value within the epoch	[20,41]
Minimum *Z* (Min*Z*)	The minimum *Z*-axis acceleration value within the epoch	[20,41]

**Table 3 animals-08-00012-t003:** Total number of 10 s mutually exclusive behaviour epochs. The number of animals for which behaviours were collected within each deployment is shown in parentheses.

Behaviour	Collar	Leg	Ear
Sound walking	95 (3)	94 (3)	274 (5)
Sound standing	106 (3)	106 (3)	862 (5)
Sound grazing	298 (4)	298 (4)	342 (5)
Sound lying	40 (1)	46 (1)	0 (0)
Lame walking	88 (3)	92 (4)	98 (4)
lame standing	62 (3)	93 (4)	97 (4)
Lame grazing	171 (3)	181 (4)	182 (4)
Lame lying	236 (3)	279 (4)	279 (4)

**Table 4 animals-08-00012-t004:** The metric order of importance for each deployment (in decreasing order of importance).

RF Variable Selection					
Analysis I			Analysis II		
Ear	Front Leg	Collar	Ear	Front Leg	Collar
MV	Ax	Ax	MV	Ax	Entropy
Ay	SMA	Az	AI	SMA	Az
Energy	Az	Entropy	Ay	AI	Max-Z
SMA	AI	AI	SMA	Max-*X*	Energy
AI	MV	Energy	Energy	Az	AI
Min-*X*	Max-*Y*	Max-*Z*	Min-*Z*	MV	MV
Max-*Y*	Max-*X*	MV	Min-*X*	Energy	Ax
Max-*X*	Max-*Z*	Max-*X*	Az	Max-*Y*	Min-*X*
Min-*Z*	Energy	Min-*Z*	Max-*Y*	Entropy	Min-*Z*
Az	Entropy	Min-*X*	Min-*Y*	Min-*X*	Max-*X*
Min-*Y*	Ay	SMA	Ax	Ay	Min-*Y*
Max-*Z*	Min-*Y*	Min-*Y*	Max-*Z*	Max-*Z*	SMA
Ax	Min-*X*	Ay	Entropy	Min-*Y*	Ay
Entropy	Min-*Z*	Max-*Y*	Max-*X*	Min-*Z*	Max-*Y*

**Table 5 animals-08-00012-t005:** QDA confusion matrices of the leave-one-out cross validation analysis for the classification of six mutually exclusive behaviours (Analysis I) using accelerometers deployed in three locations (collar, leg, and ear). The QDA algorithm used the top three ranked metrics from the respective data sets for the discrimination of behavior. Correctly predicted events are shown in **bold** and misclassifications in red.

Deployment Location	Predicted Behaviour (Events)	Observed Behaviour (Events)					
		Sound Grazing	Sound Standing	Sound Walking	Sound Lying	Lame Walking	Lame Grazing
Ear	Sound grazing	**287**	42	5		6	131
	Sound standing	44	**812**	2		3	11
	Sound walking	1	4	**241**		9	3
	Sound lying						
	Lame walking	0	0	14		**78**	3
	Lame grazing	10	3	11		2	**34**
	Prediction accuracy	84%	94%	88%		80%	19%
Collar	Sound grazing	**281**	26	0	0	0	39
	Sound standing	12	**53**	12	3	29	14
	Sound walking	0	22	**77**	13	13	2
	Sound lying	2	5	5	**22**	4	4
	Lame walking	0	0	0	1	**11**	4
	Lame grazing	3	0	0	0	20	**108**
	Prediction accuracy	94%	50%	82%	56%	14%	63%
Leg	Sound grazing	**262**	34	0	0	5	55
	Sound standing	33	**71**	0	0	6	105
	Sound walking	0	0	**81**	0	2	0
	Sound lying	2	0	0	**46**	0	8
	Lame walking	1	0	13	0	**80**	6
	Lame grazing	0	1	0	0	0	**7**
	Prediction accuracy	88%	67%	86%	100%	86%	4%

**Table 6 animals-08-00012-t006:** QDA confusion matrices of the leave-one-out cross validation analysis for the classification of five mutually exclusive behaviours (Analysis II) using accelerometers deployed in three locations (collar, leg, and ear). The QDA algorithm used the top three ranked metrics from the respective data sets for the discrimination of behaviour. Correctly predicted events are shown in **bold** and misclassifications in red.

Deployment Location	Predicted Behaviour (Events)	Observed Behaviour (Events)				
		Sound Grazing	Sound Standing	Sound Walking	Sound Lying	Lame Walking
Ear	Sound grazing	**321**	26	1		6
	Sound standing	15	**830**	1		3
	Sound walking	2	5	**262**		9
	Sound lying					
	Lame walking	4	0	9		**80**
	Prediction accuracy	94%	96%	96%		82%
Collar	Sound grazing	**283**	26	2	0	0
	Sound standing	13	**53**	10	12	9
	Sound walking	2	15	**60**	8	42
	Sound lying	0	12	23	**18**	6
	Lame walking	0	0	0	2	**31**
	Prediction accuracy	95%	50%	63%	45%	35%
Leg	Sound grazing	**266**	44	0	0	3
	Sound standing	26	**61**	0	0	6
	Sound walking	0	0	**60**	0	3
	Sound lying	0	0	0	**46**	0
	Lame walking	6	1	34	0	**81**
	Prediction accuracy	89%	58%	64%	100%	87%

**Table 7 animals-08-00012-t007:** Performance statistics of the leave-one-out cross validation for the QDA classification model to discriminate between the five mutually exclusive behaviours from Analysis II.

Deployment Location	Predicted Behaviour (Events)	Observed Behaviour (Events)				
		Sound Grazing	Sound Standing	Sound Walking	Sound Lying	Lame Walking
Ear	MV, AI, Ay					
	Sensitivity	94%	96%	96%		82%
	Specificity	97%	97%	99%		99%
	Accuracy	96%	97%	98%		98%
	Precision	91%	98%	94%		82%
Collar	Entropy, Az, Max-*Z*					
	Sensitivity	95%	50%	63%	45%	35%
	Specificity	90%	92%	87%	96%	90%
	Accuracy	92%	85%	84%	93%	83%
	Precision	91%	55%	47%	45%	35%
Leg	Ax, SMA, AI					
	Sensitivity	89%	58%	64%	100%	87%
	Specificity	83%	94%	99%	100%	98%
	Accuracy	86%	88%	94%	100%	96%
	Precision	85%	66%	95%	100%	87%

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
