# Peer review of "Predicting Lameness in Sheep Activity Using Tri-Axial Acceleration Signals"

_animals, 2018, doi:10.3390/ani8010012_

Round 1
Reviewer 1 Report
This was an interesting and useful piece of work that I would recommend to be published.
The parameters that have been used to identify lameness are similar to those that have been used in other publications but I have not seen an analysis of this detail applied to sheep.
If I was to find a criticism it would relate to the calculation of the metrics used to quantify the performance namely sensitivity, specificity, accuracy and precision. These always challenge the reader even when they have been clearly defined, as they have here. If I use the definition of sensitivity TP/(TP +FN) I can calculate values that are equal to the values quoted in the tables as Prediction Accuracy.
Eg Collar – sound grazing
TP = 283, FN = 15 giving a sensitivity of 95%
I would have calculated the Precision – TP/(TP +FP) as being (for the same instance, sound grazing collar) as TP = 283, FP = 26 +2 = 28 = 91%.
I would like to ask the authors to help the reader out and give an example of the calculation.
The analysis has been applied to an instance where the measurements are likely to be exaggerated from what might take place in a real environment due to the nature of the experiment. I can see flaws in that but I also recognise some value in that it bounds the question. The authors recognise this limitation and discuss it therefore I am content.
Overall this is a piece of work that I enjoyed reading and would be happy to see published. It will give an insight to others developing algorithms.
Reviewer 2 Report
Overview
The paper has been well written, providing a clear narrative throughout that is easy to follow. The research addresses an area that is significantly lacking for the sheep industry, but well established in other areas, which is good to see. The use of the extreme lameness technique is novel and although justification for this was provided, it is not clear whether this is representative of naturally occurring lameness. Lameness can occur in sheep at many different levels and differing number of limbs/feet affected. Application of this technique to observe naturally occurring lameness is much needed to fully justify the results. It is not clear why only 5 animals were used when 10 animals were available. Five animals is a very limited data set and raises questions with regard the validity of the results. Although it is good to reduce the number of animals used in any experiments, consideration must be made as to the minimum number of animals required to gain valid results. The paper maybe better proposed as a pilot study with further work carried out to improve the validity of the results rather than a full research paper. I would also recommend you analyse your behavioural footage as well to answer some of your other questions raised in the discussion. This would allow for a much more rounded paper. There are some areas of data missing or contradictory information, and literature key to this area have not be considered. See below for more detailed comments.
Detailed
Line 1-2 grammar: using a signals? Remove the “a”.
Line 57: give a reference for the “accelerometer”, as you have done for the other technologies.
Line 58: Expand/explain “practical considerations should be taken into account”.
Line 73: Use English or American language rather than a mixture. “Behaviour” is English, whereas “recognize” is American.
Line 90: You state previous studies but do not give any references for this.
Line 91: you mention “Controlled environment” but it is not clear what exactly you mean by this and what the link then is to line 91.
Line 94: You state “Noting the desirability of an ear-tag” but you do not give justification or reason to make this assumption.
Line 102-103: Why only 5 animals used if 10 available?
Line 104: What was the purpose of “companion” animals for this study. Did they act as controls of simply to reduce stress? Were controls not considered, or did each individual act as their own control? This needs to be made clear.
Were all sheep free from disease and pain before the study started? How had they previously been housed/managed?
Line 107: should read “to three locations on each candidate”.
Line 108: states that collars were placed in nearside, whilst ear tags on offside, yet in the picture provided (figure 1.) they are on the same side. In figure 3 however, they are on different sides.
Line 115-116: “consistent with normal ear tag deployment” – were the ear tags already present on the sheep, or did you need to ear tag for this project? Make this clear as there are ethical implications for the later.
No ethical approval information has been given. This should be included to show that ethical approval from your institution was provided and concerns were addressed.
Line 137: Why were 2 statistical packages used?
Line 162: Were animals given an adjustment period to get used to the instrumentation before observations began?
Figure 2: Were animals matched or did animals act as their own control? Were observations of sound and lame activity alternated or were all sheep exposed to the same order of activity?
Line 174 and line 199-200: Information on sampling technique for behaviours needed e.g. one-zero sampling or instantaneous sampling.
Line 175: In this section I think you need to give your justification for using this technique as you did in line 400-401.
Line 201: Were you looking for a full 10 seconds of lame walking – would this be likely given the difficulty of movement created by the bandaging?
Line 211-212: Provide reasoning for including sound standing and sound lying if no comparison could be made?
Line 220: reference for Random Forest should be here as first mention and not line 226.
Line 229-230: justification for choice of 4 and 500 in analysis required.
Line 224: presentation of formulas could be better with a consistent font size.
Line 253: “No apparent adverse effects” – how did you evaluate this? Doesn’t this contradict
Line 256: Table 3 – sound walking was only performed or seen in 1 animal? Is this representative? The level of lame lying data captured possibly indicates that these animals found it very difficult to move around and so instead of doing any walking or grazing they were more likely to simply lye down. This behaviour of lying when severely lame is more indicative of the effect and what is more lying to show up as being a problem. This needs more discussion. Consider analysing the behaviour as well as the tri-axial data.
Line 268: Figure 4 is not very clear at all. Consider separating out x, y, and z, or zooming in on an area that can clearly show the different amplitudes. In its current format this is unclear.
Line 293: remove the “-“ at the end of the sentence.
Line 302: table 6 has prediction accuracy data missing from the ear group. This meant that it was not possible for me to review the section on ears line 303-321.
Line 344 “a reduced amount of movement when grazing was evident” due to instrumented limb?
Line 352: in table 7 it was not clear why the ear section had no data for sound lying yet the other areas did.
Line 363: “other behaviours” – give some examples.
Line 370: extra space between et al., and [19] needs removing.
Line 391-392: “between lameness and lying behaviour is lacking for sheep” – your data potentially addresses this when viewing table 3. Consider analysing your behavioural footage.
Line 407: Lame sheep tend to do a lot of kneeling, especially when severely lame. Discuss this further.
Line 413: “Little information on ability of sheep farmers to identify lame animals” see Kaler and Green 2011. Animal Welfare, vol.20. page 321-328 and Kaler and Green 2008, BMC veterinary research, vol 4, issue 41, pages 1-9. doi:10.1186/1746-6148-4-4
Line 498: Journal title should be in italics not article title.
Reviewer 3 Report
I found the paper enjoyable and well written. I have a few questions:
Line 114 The ear tag looked very "pendulous" as you had taped the accelerometer onto the ear tag. Wouldn't this produce exaggerated movement when the animal bobs it's head. I wonder if a tag that is shorter as is more likely in a commercial setting would give as good results? Such a tag as you used would get ripped out pretty easily in sheep netting.
Line 176 You are mimicking 10/10 lameness by restraining the limb although I cannot see how in Line 177 you say that the leg remains straight?
Line 318 Your comment that the agitation and discomfort associated with pain could be investigated in the future. However several other causes of head flicking etc such as head fly exist and I do not think that it could be only attributed to lameness.
In your general discussion where you talk about cattle behaviours are these in housed conditions? cattle being fed a high concentrate diet may get their feed quickly and then choose to lie down due to pain but grazing sheep may be forced to keep grazing to avoid weight loss.
I agree that the changes in gait associated with hind and fore leg lameness are likely to be different.
